# Use of zebrafish to identify host responses specific to type VI secretion system mediated interbacterial antagonism

**Mollie Virgo** [1,2], **Serge Mostowy** [2]*, **Brian T. Ho** [1,3]*

**1** Institute of Structural and Molecular Biology, Department of Biological Sciences, Birkbeck College, London, United Kingdom, **2** Department of Infection Biology, London School of Hygiene and Tropical Medicine, Keppel Street, London, United Kingdom, **3** Institute of Structural and Molecular Biology, Division of Biosciences, University College London, London, United Kingdom

* serge.mostowy@lshtm.ac.uk (SM); b.ho@ucl.ac.uk (BTH)

## Abstract

Interbacterial competition is known to shape the microbial communities found in the host, however the interplay between this competition and host defense are less clear. Here, we use the zebrafish hindbrain ventricle (HBV) as an *in vivo* platform to investigate host responses to defined bacterial communities with distinct forms of interbacterial competition. We found that antibacterial activity of the type VI secretion system (T6SS) from both *Vibrio cholerae* and *Acinetobacter baylyi* can induce host inflammation and sensitize the host to infection independent of any individual effector. Chemical suppression of inflammation could resolve T6SS-dependent differences in host survival, but the mechanism by which this occurred differed between the two bacterial species. By contrast, colicin-mediated antagonism elicited by an avirulent strain of *Shigella sonnei* induced a negligible host response despite being a more potent bacterial killer, resulting in no impact on *A. baylyi* or *V. cholerae* virulence. Altogether, these results provide insight into how different modes of interbacterial competition *in vivo* affect the host in distinct ways.

**Editor:** Peggy A. Cotter, University of North Carolina at Chapel Hill Louis Round Wilson Special Collections Library: The University of North Carolina at Chapel Hill, UNITED STATES OF AMERICA

## Author summary

Interbacterial competition plays an important role in the dynamics of microbial communities, however the impact of such competition on host defenses is less clear. In this work, we use a zebrafish model to reductively investigate the host response to distinct forms of bacterial antagonism in well-defined bacterial communities. We looked at bacterial killing mediated by the type VI secretion system (T6SS) and observed that this form of bacterial antagonism resulted in prolonged inflammatory responses and an increase in host death, independent of any specific effector or bacterial species. By contrast, bacterial killing mediated by colicins, despite being significantly more efficient in eliminating sensitive bacteria, induced minimal host responses, resulting in a substantially better host outcome. Altogether, these results provide insight into the roles of different antibacterial systems that pathogens and commensals use inside their host.

**Data Availability Statement:** All relevant data are within the manuscript and its Supporting Information files.

**Funding:** MV is funded by a Bloomsbury Colleges PhD Studentship. Research in the BH laboratory is supported by MRC Grant (MR/T031131/1) to BTH. Research in the SM laboratory is supported by a Wellcome Trust Senior Research Fellowship (206444/Z/17/Z) to SM, European Research Council Consolidator Grant (772853 - ENTRAPMENT) to SM, and Wellcome Discovery Award (226644/Z/22/Z) to SM. The funders had no role in study design, data collection and analysis, decision to publish, or preparation of the manuscript.

**Competing interests:** The authors have declared that no competing interests exist.

## Introduction

Multispecies bacterial communities play critical roles in industry, agriculture, and human health. One of the factors contributing to the composition and dynamics of these communities are the interbacterial interactions that exist within them [1]. These interactions can include cooperation (e.g., adherence and nutrient mutualism) [2] and antagonism (e.g., contact-dependent delivery of toxic effectors and contact-independent secretion of antimicrobial toxins) [3]. Although modelling the contribution of bacterial interactions *in silico* is becoming increasingly common [4–7], studying a microbial community is often complicated by community heterogeneity and the idiosyncratic behaviors of the different species comprising the community. Microbial interactions *in vivo* are further complicated by the host serving as the physical support structure for the community and the source of metabolic, nutritional, and immune factors, all of which may directly or indirectly influence bacterial cell-cell interactions, community composition and population organization. As such, fully understanding these bacterial interactions requires the use of a robust *in vivo* model system, where defined bacterial populations can be introduced, and both host and bacterial dynamics can be precisely manipulated and measured.

To develop such a model, we turned to zebrafish (*Danio rerio)* larvae, which have been previously used to study bacterial cell-cell interactions in the hindbrain ventricle (HBV) [8,9]. The HBV is an enclosed, naturally sterile compartment, where injected bacterial communities do not directly interact with the commensal microbiome of the zebrafish, thereby avoiding complications arising from antibacterial activities of the resident microbiota [10,11]. The HBV can, however, still be accessed by leukocytes (macrophages, neutrophils), allowing the injected bacterial community to fully interface with host immune processes. This powerful combination of properties enables a reductive analysis of how different specific bacterial cell-cell interactions stimulate a host response and, in turn, how host responses modulate community dynamics of these bacteria. Moreover, zebrafish larvae develop rapidly, share >80% of human genes associated with diseases, and until four weeks post fertilization only possess an innate immune system [12]. These features allow investigation of host-microbe interactions independent of potentially complicating factors such as adaptive immunity or passive acquisition of antibodies from the mother [13], which reflect the history of the individual rather than its inherent properties.

One important mode of interbacterial interaction responsible for shaping microbial communities is the type VI secretion system (T6SS) [14,15]. The T6SS is a dynamic nanomachine that uses a contractile mechanism to propel a needle-like apparatus loaded with toxic effector proteins directly into adjacent 'prey' cells. T6SS-delivered effectors span a diverse range of biological functions [16–18], allowing for T6SS-mediated attacks on both eukaryotic [19,20] and prokaryotic [21,22] targets. Each antibacterial effector is typically encoded alongside a cognate immunity protein, which protects the producing bacterium from self and kin-cell intoxication [23–25].

T6SS-mediated bacterial antagonism can significantly impact the structure and diversity of bacterial populations *in vitro* [6,26]. Notably, however, T6SS-mediated killing can be orders of magnitude less effective *in vivo* compared to equivalent competitions *in vitro* [25,27,28] and can be less effective at eliminating competing bacteria than diffusible antibacterial toxins [29,30] or toxic metabolites [31,32]. Despite such limitations in killing potential, the T6SS is still widely conserved and present in isolates of nearly all enteric Gram-negative species [33]. Considering T6SS-mediated interactions *in vivo* have been shown to stimulate host responses following bacterial antagonism [34–36], we wondered whether the interface between T6SS-mediated competition and host responses could drive unique competitive advantages for

T6SS-carrying species *in vivo*. To address this question, we mechanistically explored the host response to different forms of bacterial competition involving different combinations of bacterial species in the zebrafish HBV infection model.

Consistent with previous observations in *Drosophila* and mouse models [34,35], we found that T6SS-mediated antagonism of *Vibrio cholerae* against a sensitive prey species, *Escherichia coli*, could induce strong inflammatory responses in our zebrafish larvae HBV infection model. Additionally, in zebrafish, these responses lead to significantly reduced host survival. We determined that this host response was not due to a single T6SS effector and that T6SS-mediated antagonism against *E. coli* elicited by a completely heterologous bacterial species, *Acinetobacter baylyi*, produced a similar inflammatory response that also led to reduced host survival. In contrast, another form of *in vivo* bacterial killing, colicin-mediated antagonism, did not impact host survival, and only induced a minimal inflammatory response, despite exhibiting a more potent antagonistic effect on *E. coli* growth and survival. Furthermore, direct injection of heat-killed *E. coli* lysate did not enhance *V. cholerae* pathogenicity in the absence of T6SS-mediated antagonism. Together, these data indicate that T6SS-mediated bacterial antagonism, unlike some other modes of interbacterial competition, produces host responses that result in substantially more negative health outcomes. Due to its limited killing potency, T6SS-mediated antagonism has the potential to produce continuous and long-lasting innate immune stimulation that is not possible with more potent forms of interbacterial antagonism like bacteriocins. These findings provide insight into the mechanisms of host immune activation and have implications for the development of bacterial antagonism-based antibiotic alternatives.

## Results

### *V. cholerae* T6SS antagonism towards *E. coli* prolongs the inflammatory response and reduces host survival

To study the interplay between host response and bacterial interactions, we injected an inoculating dose of bacteria into the zebrafish larvae HBV and tracked host survival, bacterial growth, and expression of a panel of host pro-inflammatory cytokines over time (Fig 1A). We began with *V. cholerae* because it has a well-studied T6SS [25,37] with known host-pathogen interactions [34–36] and has emerged as a model organism for studying how interbacterial interactions can drive pathogenesis [38]. To avoid any complications involving host-dependent regulation of the T6SS, we elected to use strain *2740–80*, which has a constitutively active T6SS [39,40] that is functional at 28.5°C (S1A Fig), the optimal growth temperature for zebrafish maintenance. The *2740–80* strain also has the benefit of having natural frame-shift mutation in the actin cross-linking domain of VgrG-1, the primary determinant of anti-eukaryotic T6SS activity [41,42]. To determine the appropriate inoculum size for zebrafish HBV infection, we injected different doses of *V. cholerae* and measured bacterial burden and host survival over 48 hours. Inoculation with at least 2000 CFU was sufficient for stable colonization through 24 hours post infection (hpi) (S1B Fig), and as with other well-established HBV infection models [9,43], host survival was dose-dependent (S1C Fig).

To determine whether the T6SS of *V. cholerae 2740–80* would directly affect the host, we infected zebrafish larvae with either wild type (WT) or a T6SS-null mutant (ΔT6SS) and tracked bacterial growth (Fig 1B) and larvae survival (Fig 1C). As expected, in the absence of any other bacterial species, there was no significant differences between WT and the ΔT6SS mutant.

We next looked to see if antagonism between *V. cholerae* and another bacterial species could induce host innate immune responses in the zebrafish model. We used a standard laboratory strain of *E. coli* K12 as the challenge strain. In addition to being sensitive to the T6SS of

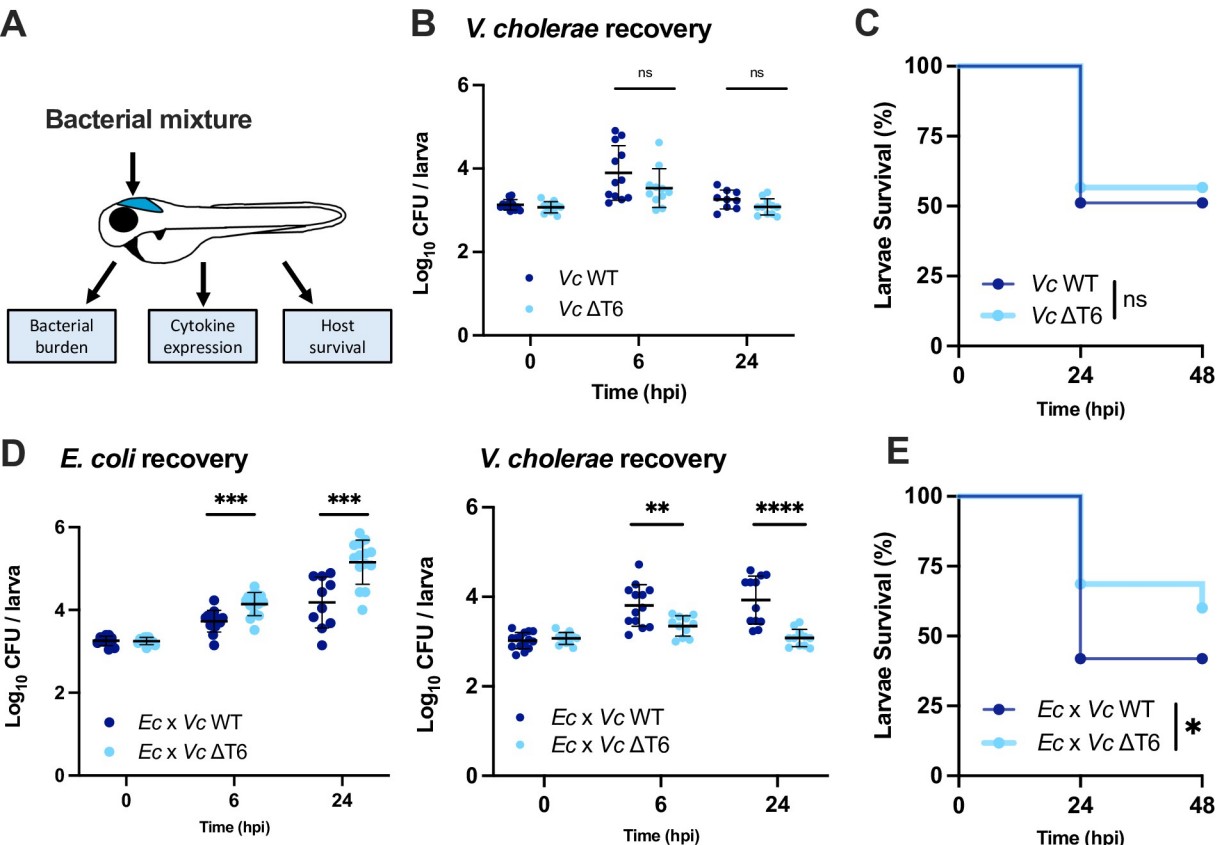

**Fig 1. V. cholerae T6SS antagonism towards E. coli reduces host survival. (A)** Illustration of zebrafish larvae indicating the site of bacterial injection (HBV) and downstream analysis workflow. **(B)** Enumeration of recovered *V. cholerae* (*Vc*) at 0, 6, or 24 hpi from larvae infected with ~2000 CFU of WT or ΔT6SS *V. cholerae*. Circles represent individual larvae. Data were pooled from 3 independent experiments, each with 3–4 larvae per time point. Significance was assessed using unpaired t-test on Log$_{10}$ transformed values. **(C)** Survival curves of larvae infected with ~2000 CFU of WT or ΔT6SS *V. cholerae*. Data were pooled from three independent experiments, each with 12–26 larvae. Significance was assessed using log-rank Mantel-Cox test. **(D)** Enumeration of recovered *E. coli* or *V. cholerae* at 0, 6, or 24 hpi from zebrafish larvae coinfected with a 1:1 mixture (~1500 CFU each) of *E. coli* and WT *V. cholerae* (*Ec* x *Vc* WT) or *E. coli* and ΔT6SS *V. cholerae* (*Ec* x *Vc* ΔT6). Circles represent individual larvae. Data were pooled from three independent experiments, each with 3–5 larvae per time point. Significance was assessed using an unpaired t-test on Log$_{10}$ transformed values. **(E)** Survival curves of larvae coinfected with a 1:1 mixture (~1500 CFU each) of either *E. coli* and WT *V. cholerae* (*Ec* x *Vc* WT) or *E. coli* and ΔT6SS *V. cholerae* (*Ec* x *Vc* ΔT6). Data were pooled from three independent experiments, each with 13–24 larvae. Significance was assessed by log-rank Mantel-Cox test. For all panels, bars indicate mean ± SEM. *p < 0.05, **p < 0.01, ***p < 0.001, ****p<0.0001, and ns (not significant).

*V. cholerae* (S1A Fig), this strain stably colonized the HBV through 24 hpi when up to 3000 CFU was injected by itself (S2A Fig) but induced only a minor inflammatory response (S2B Fig) and exhibited no lethality toward the host (S2C Fig). As such, host responses observed after coinfection with *V. cholerae* are likely attributable to the interactions between *V. cholerae* and *E. coli* rather than to direct interactions between the *E. coli* and the host.

We mixed *E. coli* with either WT or ΔT6SS strains of *V. cholerae* at a 1:1 ratio and injected the mixture into the HBV. In accordance with the mouse infection model [34], when the *V. cholerae* T6SS was functional, there was a significant reduction in the amount of recoverable *E. coli* and an increase in the amount of recoverable *V. cholerae* at both 6 and 24 hpi (Fig 1D). We also observed significantly less zebrafish survival following coinfection of *E. coli* with WT *V. cholerae* compared to coinfection of *E. coli* with the ΔT6SS mutant (Fig 1E).

We next wanted to explore the host response to *V. cholerae* in the presence and absence of T6SS-mediated bacterial antagonism. When *V. cholerae* was injected alone, mRNA levels of

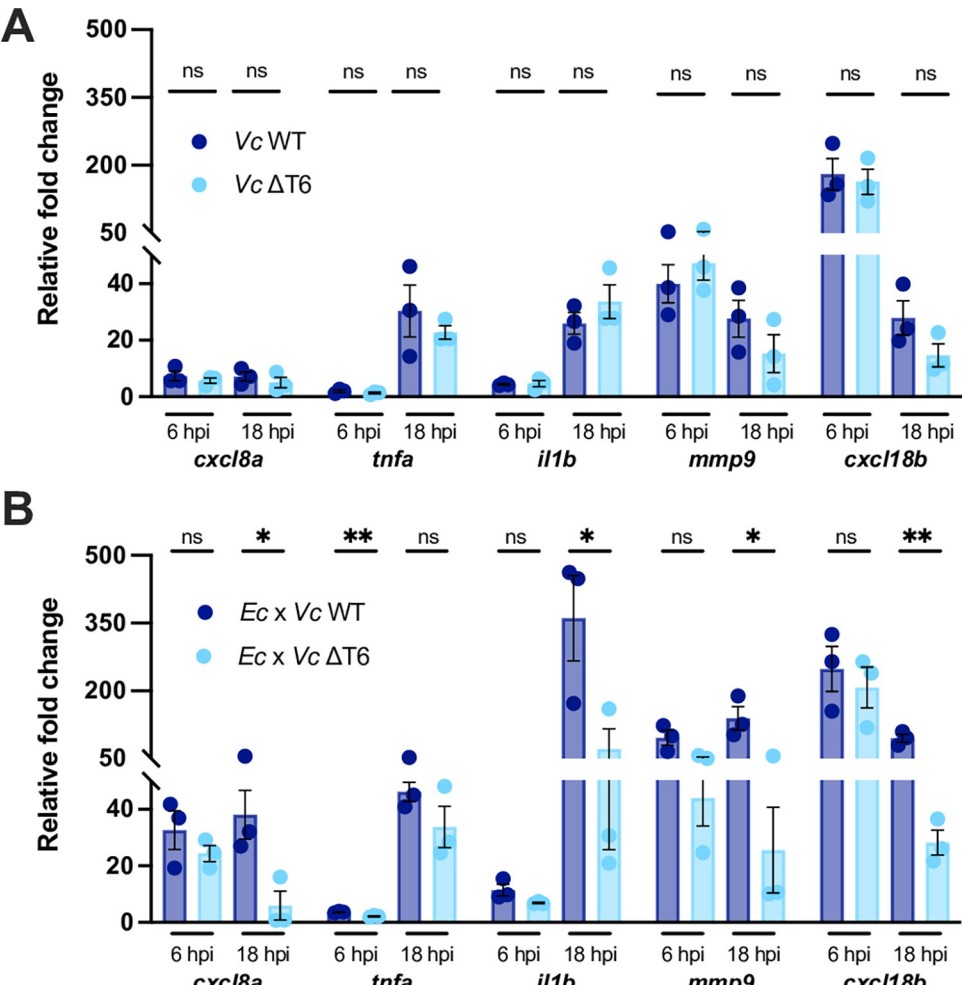

**Fig 2. *V. cholerae* T6SS-mediated antagonism towards *E. coli* induces host inflammation. (A)** Fold change in the expression of *cxcl8a*, *tnfa*, *il1b*, *mmp9* and *cxcl18b* in larvae infected with ~2000 CFU of WT or ΔT6SS *V. cholerae* relative to mock infected larvae at 6 hpi or 18 hpi. Data were pooled from three independent experiments, each with 12–15 larvae per time point. Significance was assessed using unpaired t-test on Log$_2$ transformed values. **(B)** Fold change in the expression of *cxcl8a*, *tnfa*, *il1b*, *mmp9* and *cxcl18b* in larvae coinfected with a 1:1 mixture (~1500 CFU each) of either *E. coli* and WT *V. cholerae* (*Ec* x *Vc* WT) or *E. coli* and ΔT6SS *V. cholerae* (*Ec* x *Vc* ΔT6) relative to mock infected larvae at 6 hpi or 18 hpi. Data were pooled from three independent experiments, each with 10–15 larvae per time point. Significance was assessed using unpaired t-test Log$_2$ on transformed values. For all panels, bars indicate mean ± SEM. *p < 0.05, **p < 0.01, and ns (not significant).

*tnfa* and *il1b* incrementally increased throughout infection, *mmp9* and *cxcl18b* levels were highly elevated at 6 hpi before decreasing by 18 hpi, and *cxcl8a* levels increased by 6 hpi but remained constant through 18 hpi (Fig 2A). Just as with larvae survival and *V. cholerae* burden, there were no significant differences between host innate immune responses to WT and the ΔT6SS mutant (Fig 2A).

When *V. cholerae* was injected along with *E. coli*, the host cytokine response at 6 hpi for both the WT and ΔT6SS competitions (Fig 2B) were very similar to each other and to injection of *V. cholerae* alone (Fig 2). However, by 18 hpi, there was a marked difference between the two competitions as the larvae with WT *V. cholerae* maintained elevated cytokine expression while those with the ΔT6SS mutant returned closer to baseline (Fig 2B). The overall similarity of the host responses to T6SS-mediated interbacterial competition between zebrafish and mice

serves as validation of the zebrafish HBV as a model system to further dissect the mechanisms of microbial community-host interactions.

## T6SS-mediated antagonism by *A. baylyi* also induces host inflammatory response

*V. cholerae* has three antibacterial T6SS effectors: TseL is a lipase [25]; VasX contains a colicin-like domain and has been suggested to act as a pore-forming toxin [44,45]; and VgrG-3 contains a C-terminal hydrolase domain associated with peptidoglycan degradation [46]. Consistent with previous *in vitro* studies [37], during *in vivo* competition, no individual effector is solely responsible for the T6SS-mediated antagonism between *V. cholerae* and *E. coli* (S3A Fig). During coinfection with *E. coli* in the zebrafish HBV, although the strain lacking the VasX effector seemed to colonize slightly better at 6 hpi, no individual effector conferred a significant growth advantage or produced significant differences in the expression of host inflammatory cytokines (S3B Fig) or host survival (S3C Fig). These data suggest that host inflammatory responses are occurring irrespective of the precise molecular mode of bacterial cell death and could therefore be a universal consequence of T6SS-mediated competition.

To test this hypothesis, we used the zebrafish larvae HBV model to investigate whether another bacterial species carrying a functional T6SS could also induce similar host responses. *Acinetobacter baylyi* ADP1 is a non-pathogenic, naturally competent soil bacterium encoding a single, constitutively active antibacterial T6SS which can efficiently kill *E. coli* [47] (S4A Fig). Like *V. cholerae*, the *A. baylyi* T6SS is active at 28.5°C (S4A Fig); *A. baylyi* can stably colonize the zebrafish HBV through 24 hours (S4B Fig); and an infection dose of ~3000 CFU resulted in ~65% host survival (S4C Fig). Additionally, colonization of the HBV, host survival, and host cytokine response was not dependent on *A. baylyi* having a functional T6SS (S4D, S4E and S4F Fig).

During coinfection with *E. coli*, we observed a T6SS-dependent reduction in the number of recovered *E. coli* indicating that interbacterial antagonism was occurring (Fig 3A). However, unlike *V. cholerae*, having a functional T6SS did not allow WT *A. baylyi* to colonize the HBV to a greater extent than the ΔT6SS mutant (Fig 3A). While this difference between the two species could be due to differences in growth rate *in vivo*, considering that the magnitude of cytokine response to *A. baylyi* was significantly lower than those of *V. cholerae*, it is possible that the innate immune response may be managing the growth of these species differently. Despite these differences, T6SS-mediated antagonism between *A. baylyi* and *E. coli* also caused a prolonged activation of inflammatory cytokines that were particularly apparent by 42 hpi (Fig 3B) and resulted in a substantial reduction in larvae survival by 48 hpi (Fig 3C). Altogether, these results show that host inflammation is a generalized response to T6SS-mediated antagonism and is not dependent on a particular toxic effector or specific bacterial species.

## Dampening the inflammatory response resolves T6SS-mediated differences in host survival

To further resolve how T6SS-mediated antagonism was reducing host survival, we looked to see if dampening the host inflammatory response could eliminate such T6SS-dependent differences. Dexamethasone (DEX) is a corticosteroid anti-inflammatory drug previously shown to reduce leukocyte recruitment in zebrafish [48]. It blocks vasodilation and immune cell migration through suppression of pro-inflammatory cytokines [49]. DEX was administered by immediately bathing the zebrafish larvae in water containing 50 μg/mL of the drug following bacterial injection.

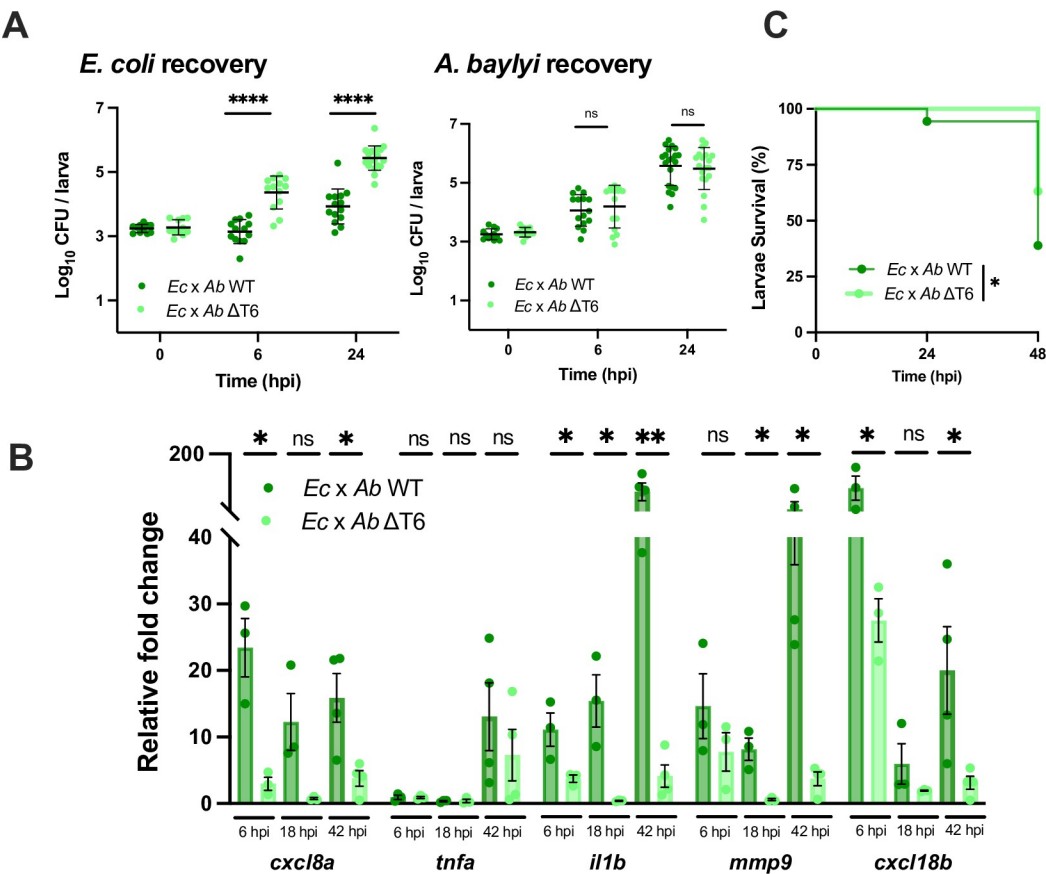

**Fig 3. *A. baylyi* T6SS-mediated antagonism towards *E. coli* induces host inflammation and reduces host survival. (A)** Enumeration of recovered *E. coli* or *A. baylyi* at 0, 6, or 24 hpi from zebrafish larvae coinfected with a 1:1 mixture (~2000 CFU each) of *E. coli* and WT *A. baylyi* (*Ec* x *Ab* WT) or *E. coli* and ΔT6SS *A. baylyi* (*Ec* x *Ab* ΔT6). Circles represent individual larvae. Data were pooled from three independent experiments, each with 3–5 larvae per time point. Significance was assessed using unpaired t-test on Log₁₀ transformed values. **(B)** Fold change in the expression of *cxcl8a*, *tnfa*, *il1b*, *mmp9* and *cxcl18b* in larvae coinfected with a 1:1 mixture (~2000 CFU each) of either *E. coli* and WT *A. baylyi* (*Ec* x *Ab* WT) or *E. coli* and ΔT6SS *A. baylyi* (*Ec* x *Ab* ΔT6SS) relative to mock infected larvae at 6 hpi, 18 hpi, or 42 hpi. Data were pooled from three independent experiments, each with 10–15 larvae per time point. Significance was assessed using unpaired t-test on Log₂ transformed values. **(C)** Survival curves of larvae coinfected with a 1:1 mixture (~2000 CFU each) of either *E. coli* and WT *A. baylyi* (*Ec* x *Ab* WT) or *E. coli* and ΔT6SS *A. baylyi* (*Ec* x *Ab* ΔT6). Data were pooled from three independent experiments, each with 10–15 larvae. Significance was assessed by log-rank Mantel-Cox test. For all panels, bars indicate mean ± SEM. *p < 0.05, ****p<0.0001, and ns (not significant).

Although DEX treatment did not affect T6SS-dependent antagonism against *E. coli* by *V. cholerae* (Fig 4A) or *A. baylyi* (Fig 4B), it did eliminate T6SS-dependent differences in host survival for both species. However, the way these differences resolved was unexpectedly different for the two species. For *V. cholerae*, DEX caused coinfection of *E. coli* and the ΔT6SS to become as lethal as the coinfection with WT (Fig 4C). Meanwhile, for *A. baylyi*, DEX increased the survival of larvae coinfected with WT to match that of the ΔT6SS strain (Fig 4D). This difference in the effect of DEX is indicative of how the host innate immune system responds differently to each species. For *V. cholerae*, in the absence of T6SS, DEX treatment resulted in a significant increase in *V. cholerae* bacterial burden (Fig 4A), indicating that host inflammation actively suppresses *V. cholerae* growth. Indeed, even in the absence of *E. coli* DEX treatment resulted in a significant increase in bacterial burden and reduction in larvae survival

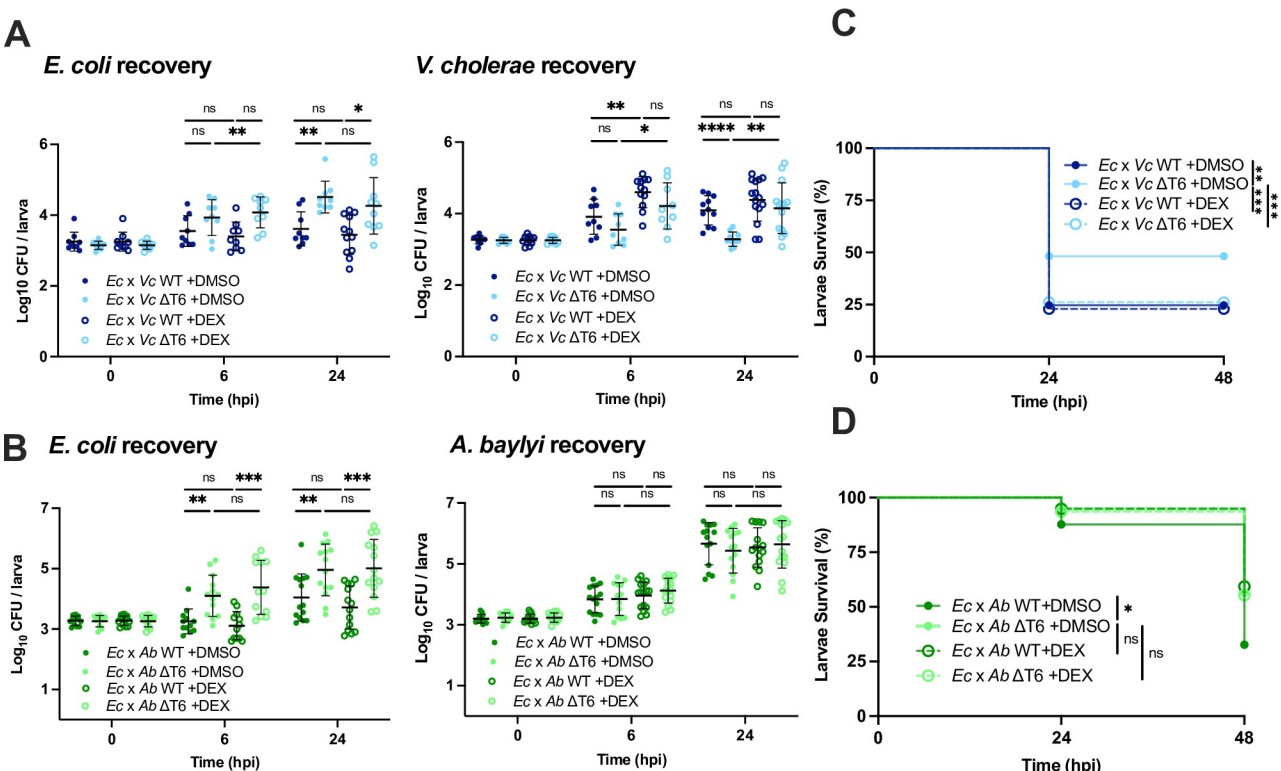

**Fig 4. Dexamethasone treatment eliminates T6SS-dependent effects on host survival. (A)** Enumeration of recovered *E. coli* or *V. cholerae* at 0, 6, or 24 hpi from zebrafish larvae coinfected with a 1:1 mixture (~1500 CFU each) of *E. coli* and WT *V. cholerae* (*Ec* x *Vc* WT) or *E. coli* and ΔT6SS *V. cholerae* (*Ec* x *Vc* ΔT6), then immersed in 50 μg/mL dexamethasone (DEX) or solvent control (DMSO). Circles represent individual larvae. Data were pooled from three independent experiments, each with 3 larvae per time point. Significance was assessed by unpaired t-test on Log$_{10}$ transformed values. **(B)** Enumeration of recovered *E. coli* or *A. baylyi* at 0, 6, or 24 hpi from larvae coinfected with a 1:1 mixture (~2000 CFU each) of *E. coli* and WT *A. baylyi* (*Ec* x *Ab* WT) or *E. coli* and ΔT6SS *A. baylyi* (*Ec* x *Ab* ΔT6), then immersed in 50 μg/mL DEX or DMSO. Circles represent individual larvae. Data were pooled from three independent experiments, each with 3 larvae per time point. Significance was assessed by unpaired t-test on Log$_{10}$ transformed values. **(C)** Survival curves of larvae coinfected with a 1:1 mixture (~1500 CFU each) of *E. coli* and WT *V. cholerae* (*Ec* x *Vc* WT) or *E. coli* and ΔT6SS *V. cholerae* (*Ec* x *Vc* ΔT6), then immersed in 50 μg/mL dexamethasone (DEX) or solvent control (DMSO). Data are pooled from four independent experiments, each with 13–29 larvae. Significance was assessed by log-rank Mantel-Cox test. **(D)** Survival curves of larvae coinfected with a 1:1 mixture (~2000 CFU each) of *E. coli* and WT *A. baylyi* (*Ec* x *Ab* WT) or *E. coli* and ΔT6SS *A. baylyi* (*Ec* x *Ab* ΔT6), then immersed in 50 μg/mL dexamethasone (DEX) or solvent control (DMSO). Data are pooled from three independent experiments, each with 10–25 larvae. Significance was assessed by log-rank Mantel-Cox test. For all panels, bars indicate mean ± SEM. *p < 0.05, **p < 0.01, ***p < 0.001, ****p<0.0001, and ns (not significant).

(S5A and S5B Fig). By contrast, DEX treatment does not noticeably affect *A. baylyi* growth in competition with *E. coli* (Fig 4B) or when injected alone (S5C Fig), and it did not affect larvae survival in the absence of *E. coli* (S5D Fig).

## Heat-killed *E. coli* lysate alone does not enhance *V. cholerae* virulence

Having observed that T6SS-mediated bacterial antagonism was facilitating *V. cholerae* growth, we next wondered whether simply adding dead *E. coli* cells could produce a similar effect in the absence of T6SS-mediated killing. We mixed WT or ΔT6SS mutant *V. cholerae* with heat-killed *E. coli* cells and injected the mixture into zebrafish larvae. In this case, we observed no significant differences in larvae survival (Fig 5A), *V. cholerae* growth (Fig 5B), or inflammatory cytokine expression (Fig 5C) between the competition with heat-killed *E. coli* (and WT or ΔT6SS mutant *V. cholerae*) versus the competition with live *E. coli* and the ΔT6SS mutant.

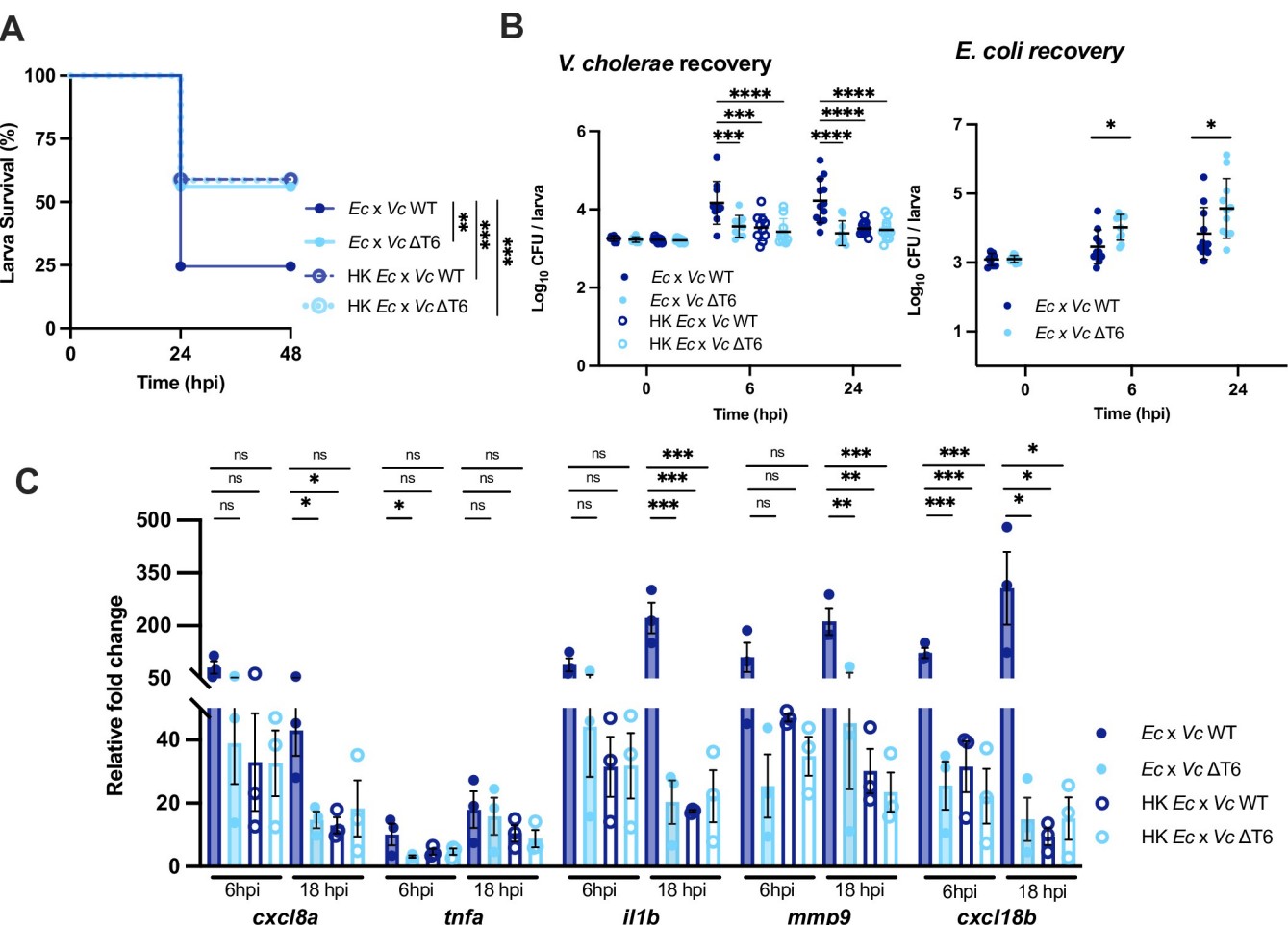

**Fig 5. Heat-killed *E. coli* lysate does not enhance *V. cholerae* pathogenicity in the absence of T6SS-mediated killing. (A)** Survival curves of larvae coinfected with a 1:1 mixture of *E. coli* (Ec) or heat-killed *E. coli* lysate (HK Ec) with WT *V. cholerae* (*Vc* WT) or ΔT6SS *V. cholerae* (*Vc* ΔT6) (~1500 CFU each). Data are pooled from three independent experiments, each with 11–20 larvae. Significance was assessed by log-rank Mantel-Cox test. **(B)** Enumeration of recovered *V. cholerae* or *E. coli* at 0, 6, or 24 hpi from zebrafish larvae coinfected with a 1:1 mixture of *E. coli* (Ec) or heat-killed *E. coli* lysate (HK Ec) with WT *V. cholerae* (*Vc* WT) or ΔT6SS *V. cholerae* (*Vc* ΔT6) (~1500 CFU each). Circles represent individual larvae. Data were pooled from three independent experiments, each with 3–5 larvae per time point. Significance was assessed by unpaired t-test on $Log_{10}$ transformed values. **(C)** Fold change in the expression of *cxcl8a*, *tnfa*, *il1b*, *mmp9* and *cxcl18b* in larvae coinfected with 1:1 mixture of *E. coli* (Ec) or heat-killed *E. coli* lysate (HK Ec) with WT *V. cholerae* (*Vc* WT) or ΔT6SS *V. cholerae* (*Vc* ΔT6) (~1500 CFU each) relative to mock infected larvae at 6 hpi or 18 hpi. Data were pooled from three independent experiments, each with 10–15 larvae per time point. Significance was assessed by one-way ANOVA with Tukey's multiple comparisons test on $Log_2$ transformed values. For all panels, bars indicate mean ± SEM. *$p < 0.05$, **$p < 0.01$, ***$p < 0.001$, ****$p<0.0001$, and ns (not significant).

## Colicin-mediated bacterial antagonism does not promote host inflammation

Having observed that antibacterial activities of both the *V. cholerae* and *A. baylyi* T6SS are able to modulate host cytokine responses, we next tested whether similar host responses could be induced by an entirely different mode of bacterial antagonism, colicins [50]. Unlike T6SS, colicins are secreted or released into the surrounding environment and do not require direct cell-cell contact. Colicins are known to be responsible for *E. coli* killing by *Shigella sonnei* and have been shown to be present in the epidemiological successful *S. sonnei* lineages, highlighting their importance in interbacterial competition [30].

S. *sonnei* 53G encoded colicin E1 (ColE1) has been shown to eliminate commensal *E. coli in vivo* [51]. Taking advantage of its well-characterized behavior in the zebrafish infection model

[52] we used an avirulent strain of S. *sonnei* 53G lacking the pINV virulence plasmid but retaining colicin production (S6 Fig) as a colicin delivery vehicle to determine the host response to colicin-mediated bacterial killing. When we coinfected the zebrafish HBV with this colicin producing strain (col+) along with *E. coli*, there was a rapid elimination of *E. coli* compared to when the colicin gene was disrupted (col−) with a corresponding increase in the amount of *S. sonnei* (Fig 6A). Despite the potency of the antibacterial activity, the presence of colicins did not elicit any difference in inflammatory cytokine expression (Fig 6B) and had no effect on host survival (Fig 6C).

Given that neither *E. coli* nor the pINV-deficient *S. sonnei* were virulent toward zebrafish larvae, we wondered if bacterial antagonism-dependent host immune responses required the presence of a pathogen to manifest. We therefore also performed these competition experiments in the presence of ΔT6SS mutant *V. cholerae* or *A. baylyi*. *E. coli* and *S. sonnei* (col+ or col−) were mixed with *V. cholerae* or *A. baylyi* at a 2:1:1 (*V. cholerae* or *A. baylyi*:*E. coli*:*S. sonnei*) ratio and coinjected into the zebrafish HBV. As before, *E. coli* was rapidly killed when the colicin was present (Fig 6D and 6E), and as expected from the narrow target range of colicins [53], *V. cholerae* and *A. baylyi* were unaffected by colicin production (Fig 6D and 6E). When *A. baylyi* was present, the col−*S. sonnei* strain exhibited a slight growth advantage compared to the colicin producer (Fig 6E), but *V. cholerae* did not have a similar effect. Neither *V. cholerae* (Fig 6F) nor *A. baylyi* (Fig 6G) exhibited any colicin-dependent differences in host survival. Collectively, these data show that colicin mediated antagonism does not exacerbate *V. cholerae* or *A. baylyi* infection like T6SS-mediated antagonism.

## Discussion

Here, we show that the zebrafish larva HBV can be used as a highly adaptable *in vivo* platform to investigate defined microbial communities and their interactions with the host, independent of the noise associated with background commensals or adaptive immune responses. We found that antibacterial activity of the T6SS can stimulate host inflammation, sensitizing the host to infection. Removal of the host inflammatory response through DEX treatment was sufficient to resolve differences in larvae survival +/- T6SS mediated antagonism. While T6SS-mediated antagonism elicited by T6SS effector mutants or different bacterial species could all stimulate an inflammatory response, colicin-mediated antagonism and injection of heat-killed bacteria did not. Taken together, we conclude that the outcome of different modes of interbacterial competition have distinct effects on the host response. These findings provide insights into how pathogenic bacteria might specifically employ a T6SS to exploit the host inflammatory response to their advantage and the detriment of the host.

Previous studies have demonstrated that release of Microbe-Associated Molecular Patterns (MAMPs) from T6SS-mediated bacterial cell lysis can induce the host inflammatory response [34]. Considering that inflammatory mediators have short half-lives [54,55], acute inflammatory responses require constant stimulation. Together with our data demonstrating differences in host response to bacterial killing due to T6SS versus colicin activity, we propose a model for how the timing and duration of bacterial lysis can shape the degree of host response (Fig 7A). Although both T6SS and colicins can lead to the rapid destruction of a targeted cell [56,57], because colicins are diffusible the efficiency of colicin-mediated killing at a population level can be significantly higher [29]. This rapid elimination of the target bacteria means that any MAMP release will necessarily occur over a very short window resulting in an essentially imperceptible inflammatory response. By contrast, T6SS is relatively inefficient in its killing, potentially even establishing "killing zones" between killer and killed bacterial populations [26,58,59]. This effectively perpetual T6SS-mediated killing results in a continuous activation

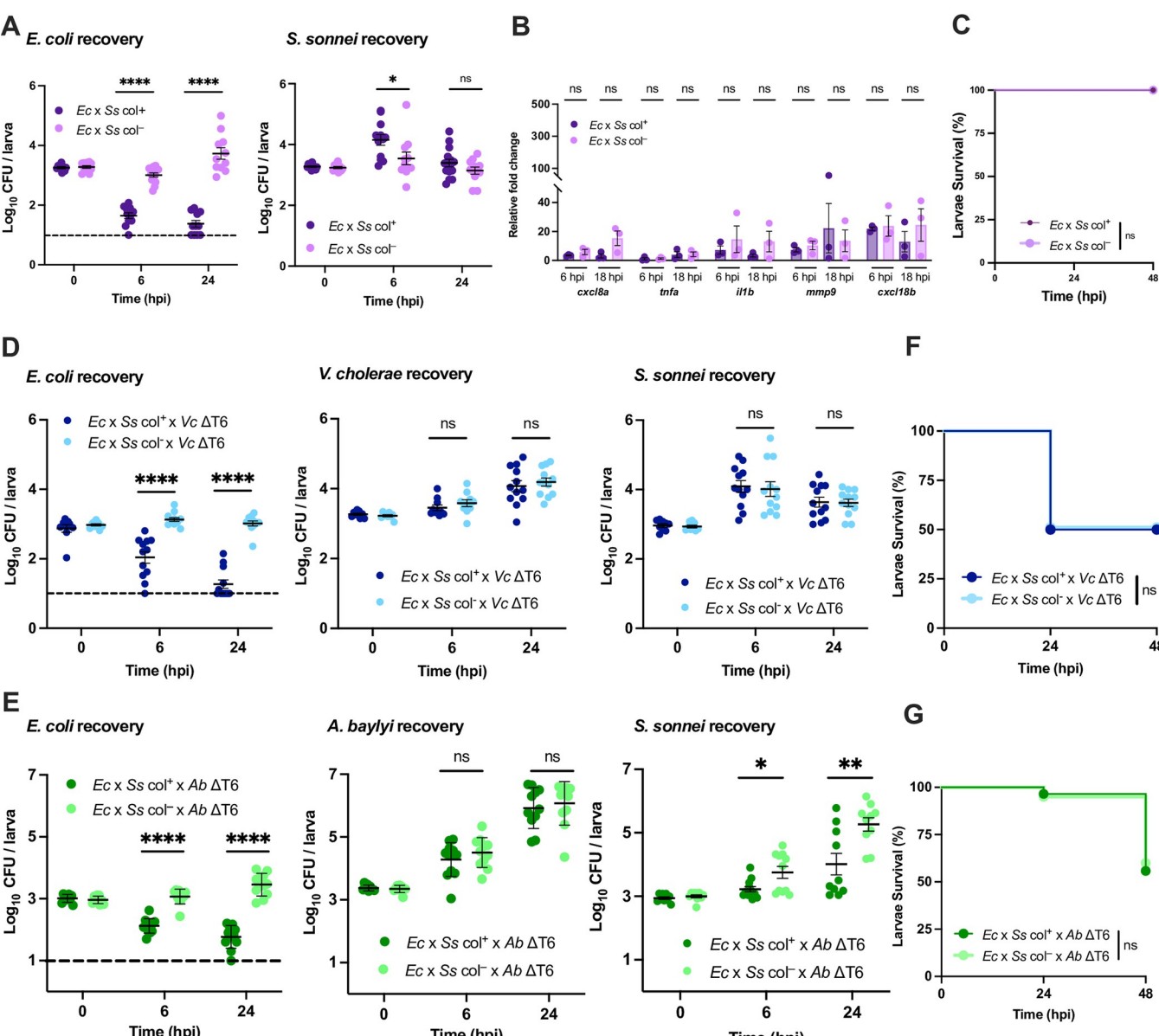

**Fig 6. Colicin-mediated antagonism does not impact host viability. (A)** Enumeration of recovered *E. coli* or *S. sonnei* at 0, 6, or 24 hpi from zebrafish larvae coinfected with a 1:1 mixture (~2000 CFU each) of *E. coli* (*Ec*) and *S. sonnei* (*Ss*) either producing (col⁺) or not producing (col⁻) colicin. Circles represent individual larvae. Data were pooled from three independent experiments, each with 3–4 larvae per time point. Significance was assessed by unpaired t-test on Log₁₀ transformed values. **(B)** Fold change in the expression of *cxcl8a*, *tnfa*, *il1b*, *mmp9* and *cxcl18b* in larvae coinfected with a 1:1 mixture (~2000 CFU each) of *E. coli* (*Ec*) and *S. sonnei* (*Ss*) either producing (col⁺) or not producing (col⁻) colicin relative to mock infected larvae at 6 hpi or 18 hpi. Data were pooled from three independent experiments, each with 10–15 larvae per time point. Significance was assessed using unpaired t-test on Log₂ transformed values. **(C)** Survival curves of larvae coinfected with a 1:1 mixture (~2000 CFU each) of *E. coli* (*Ec*) and *S. sonnei* (*Ss*) either producing (col⁺) or not producing (col⁻) colicin. There was no larvae death for either condition. Data were pooled from three independent experiments, each with 12–14 larvae. Significance was assessed by log-rank Mantel-Cox test. **(D)** Enumeration of recovered *E. coli*, *S. sonnei* and *V. cholerae* at 0, 6, or 24 hpi from larvae coinfected with a 1:1:2 mixture of *E. coli* (*Ec*, ~750 CFU), *S. sonnei* (*Ss*, ~750 CFU) either producing (col⁺) or not producing (col⁻) colicin, and T6SS-defective *V. cholerae* (*Vc* ΔT6, ~1500 CFU). Circles represent individual larvae. Data were pooled from three independent experiments, each with 3–4 larvae per time point. Significance was assessed by unpaired t-test on Log₁₀ transformed values. **(E)** Enumeration of recovered *E. coli*, *S. sonnei* and *A. baylyi* at 0, 6, or 24 hpi from larvae coinfected with a 1:1:2 mixture of *E. coli* (*Ec*, ~1000 CFU), *S. sonnei* (*Ss*, ~1000 CFU) either producing (col⁺) or not producing (col⁻) colicin, and T6SS-defective *A. baylyi* (*Ab* ΔT6, ~2000 CFU). Circles represent individual larvae. Data were pooled from three independent experiments, each with 3–4 larvae per time point. Significance was assessed by unpaired t-test on Log₁₀ transformed values. **(F)** Survival curves of larvae coinfected with a 1:1:2 mixture of *E. coli* (*Ec*, ~750 CFU), *S. sonnei* (*Ss*, ~750 CFU) either producing (col⁺) or not producing (col⁻) colicin, and ΔT6SS *V. cholerae* (*Vc* ΔT6, ~1500 CFU). Data were pooled from three independent experiments, each with 18–25 larvae. Significance was assessed by log-rank Mantel-Cox test. **(G)** Survival curves of larvae coinfected with a 1:1:2 mixture of *E. coli* (*Ec*, ~1000 CFU), *S. sonnei* (*Ss*, ~1000 CFU) either producing (col⁺) or not producing (col⁻) colicin, and ΔT6SS *A. baylyi* (*Ab* ΔT6, ~2000 CFU). Data were pooled from three independent experiments, each with 18–25 larvae. Significance was assessed by log-rank Mantel-Cox test. The dashed lines in A, D, and E, indicate the

limit of detection of the assay. When no bacteria were recovered, the cell count was assigned this value for statistical analysis. For all panels, bars indicate mean ± SEM. *$p < 0.05$, **$p < 0.01$, ****$p<0.0001$, and ns (not significant).

of host responses, which can either impair its normal function in limiting bacterial growth, as was observed with *V. cholerae*, or can induce inflammation-dependent toxicity, as was observed with *A. baylyi*.

This ability to modulate the efficacy of host responses also addresses a long-standing question of why the T6SS is so prevalent among pathogenic enteric bacteria [33], despite being relatively inefficient at clearing competing bacteria from colonization niches [27,28,60]. In other words, the inefficiency of the T6SS as an antibacterial weapon might not be a limitation, but rather an advantage for the pathogen.

Additionally, considering that host responses to T6SS-mediated bacterial killing are not dependent on a specific effector or bacterial species, T6SS activity in commensal bacteria are likely subject to similar inflammatory responses. They too must weigh the benefits of suppressing bacterial competitor growth with minimizing adverse host responses. Consistent with this notion, metagenomic analysis observed more effective T6SS-mediated competition among *Bacteroides fragilis* populations in infants compared to those in adults [61] and *B. fragilis* in the human gut has been observed to repeatedly acquire inactivating mutations in its T6SS [15]. It is possible that this selective pressure to reduce or eliminate continuous T6SS activity may reflect the need for the commensal microbial population to limit its stimulation of host immune responses.

Pathogens, on the other hand, experience a distinct set of selective pressures. To proliferate, they need to establish a colonization niche and avoid host defenses. While speculation on the role of T6SS and other contact-dependent modes of bacterial antagonism during infection have leaned toward invading and displacing already established bacterial populations [27,29,34], our results suggest that although T6SS activity induces a strong inflammatory

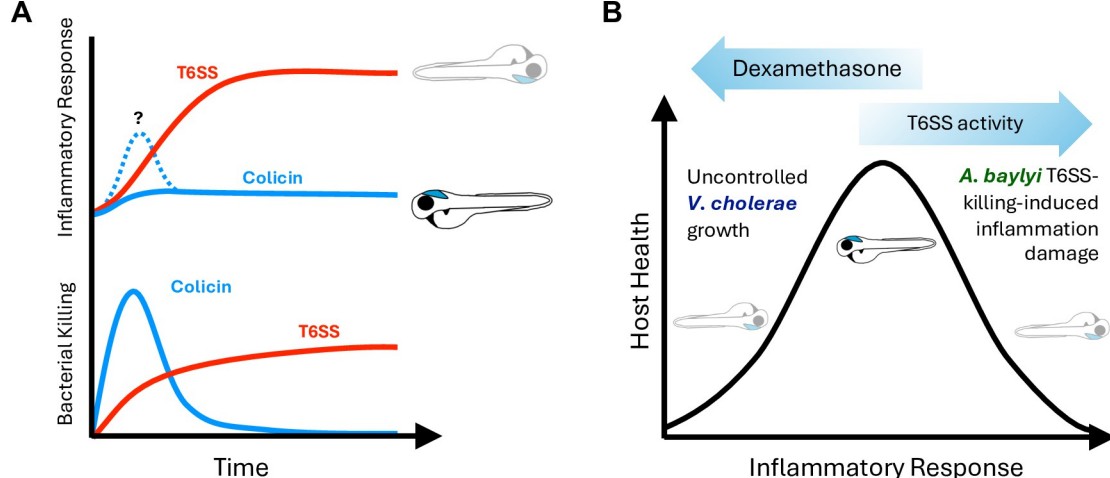

**Fig 7. Model for interbacterial antagonism induced inflammation and impact on host survival. (A)** Colicin-mediated killing is very effective at eliminating targeted bacteria. Because this killing is so rapid, if there are host inflammatory responses to the killed bacteria, they are short-lived, enabling the host to fully recover. T6SS is less efficient and therefore results in continuous bacterial killing and prolonged inflammatory response. **(B)** Health requires a balanced inflammatory response: too little enables uncontrolled pathogen growth, but too much results in host cellular damage. T6SS-mediated bacterial killing by different bacterial species impacts different parts of this scale. On one end, T6SS-induced inflammation sensitizes the host to less inherently pathogenic bacteria like *A. baylyi*, while on the other, suppression of inflammation by corticosteroids like dexamethasone allows pathogens like *V. cholerae* to grow uncontrolled. T6SS-mediated killing by *V. cholerae* exist in a unique place where the heightened inflammatory response it induces counterintuitively also enables it to grow unchecked.

response, for *V. cholerae* at least, this response is associated with an impairment in the ability of the host to clear the pathogen allowing for increased bacterial burden. That interbacterial competition can affect host health and modulate host responses in different ways highlights the fact that microbial communities are not closed systems. Evolutionary models looking to explain fitness advantages conferred by interspecies competition must therefore also factor in the responses of the host in which the microbial communities reside.

Indeed, by using the zebrafish infection model, we could directly modulate these host responses to uncover mechanistic differences in the effects of T6SS-mediated antagonism of different species. For *A. baylyi*, prolonged inflammatory response to T6SS-mediated bacterial killing is detrimental to the host. Therefore, administration of DEX to ablate host inflammation improved host survival, analogous to how DEX can be used to treat autoimmune conditions and severe cases of COVID-19 [49] or as part of adjunctive therapies for treating tuberculosis [62]. By contrast, for *V. cholerae*, T6SS-mediated bacterial killing interferes with the host's ability to clear the pathogen, meaning that self-harming inflammatory effects do not have a chance to manifest. As such, the effect of DEX is only visible in ΔT6SS mutants of *V. cholerae* or in the absence of bacteria sensitive to being killed by T6SS, where suppression of host inflammation allows *V. cholerae* to grow as efficiently as when T6SS-mediated antagonism was occurring.

Although it is not known mechanistically how T6SS-mediated antagonism promotes *V. cholerae in vivo* growth, previous studies have demonstrated that it can induce changes in the expression of *V. cholerae* virulence factors [34]. Meanwhile, another study has demonstrated that T6SS-mediated antagonism towards commensals by *V. cholerae* inhibits epithelial repair in *Drosophila* [36,63]. Furthermore, although inflammation is typically seen as a key host response that is central to helping clear infection [64], some pathogens, such as *Salmonella* Typhimurium, exploit this response to aid colonization [65]. In this instance, inflammation creates a niche which is more favourable to colonization. While the results of our DEX experiments suggest that inflammation may be important for managing *V. cholerae* infection, it also appears as though *V. cholerae* may be using host inflammation to enhance its growth. Given that this is different from what we observe with *A. baylyi*, we believe that there is likely something inherent to *V. cholerae* that is contributing towards its increase in pathogenicity in the context of host inflammation. It has been recently shown that *V. cholerae* can establish biofilm growth on the surface of immune cells as a form of predation [66]. It is possible that *V. cholerae* has evolved a strategy of using T6SS-mediated bacterial antagonism to stimulate inflammatory responses and immune cell recruitment to facilitate this form of growth. Ultimately, future work determining the spatial localization of T6SS-mediated killing and host immune cell activity during infection will go a long way toward resolving these unknowns.

On the other hand, for bacteria like *A. baylyi*, where inflammation does not significantly impact bacterial growth, DEX treatment can protect the host by preventing damage from prolonged activation of the inflammatory response (Fig 7B). Over a decade of work has shown that susceptibility to *Mycobacterium marinum* (during zebrafish infection) and *Mycobacterium tuberculosis* (during human infection) can result from either inadequate or excessive inflammation [67,68]. These studies using mycobacteria highlight how, depending on the context, inflammation can be beneficial or detrimental to the host. Our results suggest that these principles can be broadly applied to other bacterial pathogens or microbial communities, and have important implications for potential therapeutic applications of microbial communities, such as engineered microbiomes [69,70] or bacterial antagonism-based antibiotic alternatives [71,72]. Properly characterizing the behavior of such communities *in vivo* will therefore be necessary to balance controlling bacterial growth and preventing harmful inflammatory overstimulation.

## Methods

### Ethics statement

Animal experiments were performed according to the Animals (Scientific Procedures) Act 1986 and approved by the Home Office (Project licenses: PPL P4E664E3C and PP5900632). Each project license was reviewed and approved by the Animal Welfare and Ethical Review Body (AWERB) at the London School of Hygiene and Tropical Medicine (LSHTM). All experiments were conducted up to 5 dpf.

### Bacterial strains and growth conditions

A detailed strain list used in this study can be found in S1 Table. *V. cholerae* was cultured in Lennox formulation of Luria-Bertani (LB) broth or agar. *A. baylyi* strains were cultured in Terrific Broth (TB). *S. sonnei* and *E. coli* strains were grown in Tryptic Soy Broth (TSB) or Tryptic Soy Agar (TSA). Antibiotic concentrations used were: kanamycin (50μg/ml), streptomycin (50μg/ml), carbenicillin (100μg/ml), chloramphenicol (15μg/mL).

### Zebrafish husbandry

Embryos were obtained from naturally spawning zebrafish and maintained at 28.5°C in 0.5% E2 medium supplemented with 0.3 μg/mL methylene blue. Wild type AB strain zebrafish were used. Larvae were injected 3 days post fertilization (dpf) were anaesthetized with 200 μg/ml tricaine in 0.5x E2 medium. After checking for full recovery from the anesthetic, larvae were kept in E2 medium for 48hr at 28.5°C.

### Bacterial strain construction

Single effector knockouts of *2740–80* were constructed as previously described [73] using the suicide vectors pWM91-Δ*vgrG3* [25], pDS132-Δ*vasX* and pDS132-Δ*tseL* [37]. Successful deletion of target genes was confirmed using primers detailed in S2 Table. Detailed list of plasmid constructs is included in S3 Table.

Virulence plasmid pINV was removed from colicin-producing strain *S. sonnei* 53G as based on previous studies of natural pINV loss [74,75]. Bacteria (pINV+) were grown on TSA plates supplemented with 0.01% Congo red. Isolated white colonies were picked and tested by colony PCR using primers detailed in S2 Table for the absence of pINV encoded genes (*mxiG*, *ipaH1.4* and *icsB*) and presence of the colicin plasmid.

The colicin gene (*cea*) in *S. sonnei* was disrupted using lambda red recombineering using previously described protocols [76]. The *kanR-parE* cassette was amplified from pKD267 using COL_DEL_F and COL_DEL_R and inserted into the colicin operon. Successful integration was confirmed by PCR using external *cea* primers COL_EXT_F and COL_EXT_R.

Antibiotic resistance markers were added to bacterial strains for selection after competition. Since plasmid-based resistance in *V. cholerae* was lost at a high frequency, Kanamycin resistance was introduced into *2740–80* WT and ΔT6SS strains through transposon insertion using mariner transposon TnFGL3 as previously described [77]. Transposon insertion locations were determined by amplifying the insertion junctions using two-round semi-arbitrary PCR and sequencing using primers detailed in S2 Table. Independent transposon disruptions of *VC1520* were obtained for both WT and ΔT6SS strains. *VC1520* encodes an ABC-F family ATPase and was neutral for colonization in previous studies [78]. *A. baylyi* WT and *S. sonnei* strains were transformed with pMMB67EH [79] and p*rpsM-GFP* [80], respectively, to confer carbenicillin resistance. *E. coli* was transformed with pBAD33-mNeonGreen to confer chloramphenicol resistance.

### *In vitro* bacterial competition assays

T6SS killing assays were performed as previously described [21,25]. Overnight cultures were diluted 100-fold in fresh media and grown to mid-log phase. Cells were pelleted, washed in fresh media, and resuspend to $OD_{600} \approx 10$. Killer and prey cells were mixed at a 1:1 ratio (*A. baylyi* vs *E. coli*) or 5:1 ratio (*V. cholerae* vs *E. coli*) and 5 μL of the competition mix was spotted onto LB agar plates and incubated for 3 hrs at the specified temperature. Competitions were then collected, serially diluted, and spotted on agar selecting for each strain to count surviving CFU. Experiments were performed in triplicate.

   Colicin killing assays were performed as described [30] using *E. coli* NEB 10-beta as the prey and *S. sonnei* 53G pINV⁻as the predator. Overnight cultures were diluted 100-fold in fresh media and grown to mid-log phase. 10 μl of competition mixture containing a 1:10 ratio of *E. coli* to *S. sonnei* was spotted onto a pre-warmed TSA plate. Plates were incubated overnight at 28.5˚C. The competition mixture was then resuspended, serially diluted, and plated on selective media for CFU enumeration. Each experiment comprised three biological replicates.

### Zebrafish injections

Infection methods were adapted from previous protocols [43,52]. For infection inoculum preparation, overnight cultures were sub-cultured 50-fold in fresh media supplemented with appropriate antibiotics and grown to mid-log phase. Bacteria were then harvested by centrifugation, washed in phosphate buffered saline (PBS) and resuspended to the required CFU/mL in 4% polyvinylpyrrolidone (PVP) and 0.5% phenol red. For coinfections, bacterial suspensions were mixed immediately before injection at a 1:1 ratio (1500 CFU, 1500 CFU for *V. cholerae* and *E. coli*, 2000 CFU: 2000 CFU for *A. baylyi* and *E. coli* and *S. sonnei* and *E. coli*) or a 1:1:2 ratio (1000 CFU,1000 CFU:2000 CFU for *S. sonnei*, *E. coli*, and *A. baylyi*), as indicated in figure legends. For preparation of heat-killed *E. coli*, the *E. coli* inoculum was heated at 95˚C for 10 minutes prior to injection. Plating at 0 hpi confirmed that no living *E. coli* were injected. 5μL of bacterial suspension (or PBS control buffer for mock infections) was loaded into a glass capillary needle, which was manually opened to inject 0.8–1 nL into the hindbrain ventricle (HBV) of anesthetized zebrafish larvae. Injected embryos were then transferred into individual wells of a plate containing E2 medium and incubated at 28.5˚C.

### Quantification of bacterial burden and survival analysis

To determine injection inoculum or bacterial burden at each timepoint, larvae at different points post injection were mechanically homogenized in 200 μL of 0.4% Triton X-100, serially diluted, and plated onto selective plates and incubated at 37˚C. Only larvae having survived the infection were included in the analyses. For survival assays, larvae were imaged using a light stereomicroscope at 24 and 48 hpi. Larvae failing to produce a heartbeat or in which bacteria had compromised the HBV were considered nonviable. Each experiment comprised three or four biological replicates as detailed in the figure legends. Raw data values for larvae survival can be found in S1 File.

### Zebrafish chemical treatments

For suppression of the inflammatory response, immediately after infection, larvae were placed in E2 medium supplemented with 50 μg/mL dexamethasone (DEX). Control larvae were kept in E2 with 0.2% dimethyl sulfoxide (DMSO). Larvae mock infected with PBS only were also immersed in E2+ 50μg/mL DEX to ensure no adverse effects of DEX treatment on larvae survival.

### RNA extraction, cDNA synthesis and qRT-PCR

RNA was extracted from 10–15 snap-frozen larvae with the RNeasy Mini kit and reverse-transcribed using QuantiTect Reverse Transcription kit according to manufacturer's instructions. Quantitative PCR (qPCR) was performed using 7500 Fast Real-Time PCR System machine and 7500 Fast Real-Time PCR software v2.3 and SYBR green master mix. Template cDNA was subjected to PCR using primers previously described [81] and detailed in S2 Table. Each experiment comprised three biological replicates.

### Statistical analysis

Statistical tests were performed using GraphPad Prism v9.5.1 software. Data are represented as the mean ± standard errors of the mean (SEM). For larvae survival, results are plotted as Kaplan-Meier survival curves and significance determined using the log-rank (Mantel-Cox) test. Data from bacterial burden and gene expression levels were $log_{10}$- or $log_2$-transformed, respectively. Pairwise comparisons were determined using unpaired t-test on $log_{10}$- or $log_2$-transformed values as indicated in the figure legend. For multiple comparisons, one-way analysis of variance (ANOVA) test with Tukey's multiple comparisons test was used, as indicated in the figure legend. For purposes of statistical analyses, when no colonies were recovered, CFU counts were assigned as 1.

### Supporting information

**S1 Fig. Dose-dependent colonization of the zebrafish HBV by V. cholerae. (A)** *E. coli* recovery after *in vitro* competition with WT or ΔT6SS *V. cholerae* at 28.5˚C and 37˚C. Three biological replicates were performed. Significance was assessed by unpaired t-test performed on $Log_{10}$ transformed values Bars represent mean ± SEM. *p<0.05 and ns (not significant). **(B)** Enumeration of recovered *V. cholerae* at 0, 6, or 24 hpi from zebrafish larvae infected with ~750 CFU, ~2000 CFU, or ~3000 CFU of WT *V. cholerae*. Circles represent individual larvae. Data were pooled from three independent experiments, each with 3–5 larvae per time point. Bars represent mean ± SEM. **(C)** Survival curves of zebrafish larvae infected with ~750 CFU, ~2000 CFU, or ~3000 CFU of WT *V. cholerae*. Data were pooled from three independent experiments, each with 10–16 larvae.
(TIF)

**S2 Fig. Zebrafish larvae HBV infection with E. coli alone induces minimal inflammatory response. (A)** Enumeration of recovered *E. coli* at 0, 6, or 24 hpi from zebrafish larvae infected with ~3000 CFU *E. coli*. Circles represent individual larvae. Data were pooled from three independent experiments, each with 4 larvae per time point. **(B)** Fold change in the expression of *cxcl8a*, *tnfa*, *il1b*, *mmp9* and *cxcl18b* in larvae infected with ~3000 CFU *E. coli* relative to mock infected larvae at 6 hpi or 18 hpi. Data were pooled from three independent experiments, each with 10–14 larvae per time point. Significance was assessed by unpaired t-test performed on $Log_2$ transformed values. **(C)** Survival curve of zebrafish larvae infected with ~3000 CFU *E. coli*. Data were pooled from three independent experiments, each with 12 larvae. For all panels, bars indicate mean ± SEM. *p < 0.05, **p < 0.001, and ns (not significant).
(TIF)

**S3 Fig. T6SS-dependent induction of inflammation does not require any single effector. (A)** Enumeration of recovered *E. coli* or *V. cholerae* at 0, 6, or 24 hpi from zebrafish larvae coinfected with a 1:1 mixture (~1500 CFU each) of *E. coli* and WT, ΔvgrG3, ΔtseL, or ΔvasX *V. cholerae* (*Vc*) mutants. Circles represent individual larva. Data were pooled from three independent experiments each with 3–4 larvae per time point. Significance was assessed by

unpaired t-test on $Log_{10}$ transformed values. **(B)** Fold change in the expression of *cxcl8a*, *tnfa*, *il1b*, *mmp9* and *cxcl18b* in larvae coinfected with a 1:1 mixture (~1500 CFU each) of *E. coli* and WT, Δ*vgrG3*, Δ*tseL*, or Δ*vasX* *V. cholerae* (*Vc*) mutants at 6 hpi or 18 hpi. Data were pooled from three independent experiments. Significance was assessed by one-way ANOVA with Tukey's multiple comparisons test on $Log_2$ transformed values. **(C)** Survival curves of larvae coinfected with a 1:1 mixture (~1500 CFU each) of *E. coli* and WT, Δ*vgrG3*, Δ*tseL*, or Δ*vasX* *V. cholerae* (*Vc*) mutants. Data were pooled from three independent experiments, each with 15–23 larvae. Significance was assessed by log-rank Mantel-Cox test. For all panels, bars indicate mean ± SEM. \*p<0.05 and ns (not significant).
(TIF)

**S4 Fig. Dose-dependent colonization of the zebrafish HBV by A. baylyi does not require a functional T6SS. (A)** *E. coli* recovery after *in vitro* competition with WT or ΔT6SS *A. baylyi* at 28.5°C and 37°C. Three biological replicates were performed. Significance was assessed by unpaired t-test performed on $Log_{10}$ transformed values. **(B)** Enumeration of recovered *A. baylyi* at 0, 6, or 24 hpi from zebrafish larvae infected with ~750 CFU, ~3000 CFU, or ~5000 CFU of WT *A. baylyi*. Circles represent individual larvae. Data were pooled from three independent experiments, each with 3–5 larvae per time point. **(C)** Survival curves of zebrafish larvae infected with ~750 CFU, ~3000 CFU, or ~5000 CFU of WT *A. baylyi*. Data were pooled from three independent experiments, each with 10–19 larvae. **(D)** Enumeration of recovered *A. baylyi* (*Ab*) at 0, 6, or 24 hpi from larvae infected with ~2000 CFU of WT or ΔT6SS *A. baylyi*. Circles represent individual larvae. Data were pooled from 3 independent experiments, each with 3–4 larvae per time point. Significance was assessed using unpaired t-test on $Log_{10}$-transformed values. **(E)** Survival curves of larvae infected with ~2000 CFU of WT or ΔT6SS *A. baylyi*. Data were pooled from three independent experiments, each with 15–21 larvae. Significance was assessed using log-rank Mantel-Cox test. **(F)** Fold change in the expression of *cxcl8a*, *tnfa*, *il1b*, *mmp9* and *cxcl18b* in larvae infected with ~3000 CFU of WT or ΔT6SS *A. baylyi* relative to mock infected larvae at 6 hpi, 18 hpi, or 42 hpi. Data were pooled from three independent experiments, each with 10–15 larvae per time point. Significance was assessed using unpaired t-test. For all panels, bars indicate mean ± SEM. \*\*p < 0.01, and ns (not significant).
(TIF)

**S5 Fig. Dampening the host immune response causes V. cholerae, but not A. baylyi, to become more virulent. (A)** Enumeration of recovered *V. cholerae* at 0, 6, or 24 hpi from zebrafish larvae infected with a medium dose (2000 CFU) of WT *V. cholerae* or ΔT6SS *V. cholerae* (*Vc* ΔT6), then immersed in 50 μg/mL dexamethasone (DEX) or solvent control (DMSO). Circles represent individual larvae. Data were pooled from three independent experiments, each with 3–5 larvae per time point. Significance was assessed by unpaired t-test on $Log_{10}$ transformed values. **(B)** Survival curves of larvae infected with a medium dose (2000 CFU) of WT *V. cholerae* or ΔT6SS *V. cholerae* (*Vc* ΔT6), then immersed in 50 μg/mL DEX or DMSO. Data are pooled from three independent experiments, each with 12–18 larvae. Significance was assessed by log-rank Mantel-Cox test. **(C)** Enumeration of recovered *A. baylyi* at 0, 6, or 24 hpi from larvae infected with a medium dose (2000 CFU) of WT *A. baylyi* (*Ab* WT) or ΔT6SS *A. baylyi* (*Ab* ΔT6), then immersed in 50 μg/mL DEX or DMSO. Circles represent individual larvae. Data were pooled from three independent experiments, each with 3–4 larvae per time point. Significance was assessed by unpaired t-test on $Log_{10}$ transformed values. **(D)** Survival curves of larvae coinfected with a medium dose (2000 CFU) of WT *A. baylyi* (*Ab* WT) or ΔT6SS *A. baylyi* (*Ab* ΔT6), then immersed in 50 μg/mL DEX or DMSO. Data are pooled from three independent experiments, each with 10–16 larvae. Significance was assessed by log-rank

Mantel-Cox test. For all panels, bars indicate mean ± SEM. $^*$p $< 0.05$, $^{**}$p $< 0.01$, $^{***}$p $< 0.001$, $^{****}$p$<0.0001$, and ns (not significant).
(TIF)

**S6 Fig. S. sonnei shows colicin mediated antagonism towards E. coli in vitro.** *E. coli* recovery after *in vitro* competition at 28.5˚C with *S. sonnei* producing (col$^+$) or not producing (col$^-$) colicin. Three biological replicates were performed. Significance was assessed by unpaired t-test performed on $Log_{10}$ transformed values. Bars represent mean ± SEM. $^{**}$p$<0.001$.
(TIF)

**S1 File. Raw survival curve data showing the percentage survival at each time point for each experiment is shown.**
(DOCX)

**S1 Table. Strains used in this work.**
(DOCX)

**S2 Table. Primers used in this work.**
(DOCX)

**S3 Table. Plasmids used in this work.**
(DOCX)

## Acknowledgments

We thank Ho and Mostowy lab members for helpful discussions. We thank the LSHTM Biological Services Facility for their work and care of zebrafish stocks.

## Author Contributions

**Conceptualization:** Mollie Virgo, Serge Mostowy, Brian T. Ho.

**Formal analysis:** Mollie Virgo, Serge Mostowy, Brian T. Ho.

**Funding acquisition:** Serge Mostowy, Brian T. Ho.

**Investigation:** Mollie Virgo.

**Methodology:** Mollie Virgo, Serge Mostowy, Brian T. Ho.

**Project administration:** Mollie Virgo, Serge Mostowy, Brian T. Ho.

**Resources:** Serge Mostowy, Brian T. Ho.

**Software:** Mollie Virgo, Serge Mostowy, Brian T. Ho.

**Supervision:** Serge Mostowy, Brian T. Ho.

**Validation:** Mollie Virgo, Serge Mostowy, Brian T. Ho.

**Visualization:** Mollie Virgo, Serge Mostowy, Brian T. Ho.

**Writing – original draft:** Mollie Virgo, Serge Mostowy, Brian T. Ho.

**Writing – review & editing:** Mollie Virgo, Serge Mostowy, Brian T. Ho.

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
