## [Decision Letter · Decision Letter 0]

1 Apr 2024

Dear Prof Mostowy,

Thank you very much for submitting your manuscript "Type VI secretion system mediated bacterial antagonism and colicin mediated bacterial killing elicit distinct host responses and health outcomes" for consideration at PLOS Pathogens. Your manuscript was reviewed by members of the editorial board and by two independent reviewers. Both found the topic and experiments to be quite interesting, but both identified some weaknesses, including some important control experiments, that must be included in order for definitive conclusions to be drawn. These experiments, as well as other concerns, are spelled out clearly in the reviewers' reports and so I won't reiterate them here. I encourage you to consider the reviewers' comments carefully and to revise your manuscript accordingly, then to resubmit.

Please also address these comments from reviewer #2 that were apparently inadvertently added to the wrong section of the review form:

General requests:

1) Please show total n values for all survival curves. In your methods you say that each experiment represents several independent pools of 10-15 fish – while this seems appropriate, please display the actual total number of fish that comprise each survival experiment either on the graph or in the figure legend.

2) I strong suggest changing the format of your survival curves. Since you are only showing three timepoints (0h, 24h and 48h), Kaplan-Meier curves are not necessary and make your results significantly harder to read and interpret.

3) None of your experiments clearly include injection controls. Please briefly address how you controlled for off target injection-based lethality within each experiment (or include that data if you have it).

4) Figure 3 is inconclusive and belongs in the supplement. If the T6SS were employed in the absence of any effector mechanism, would perhaps the same effects on host response be observed? If the authors performed a combined knockout of the three effectors, then perhaps this would merit this figure belonging in the main text. This may be beyond the scope of this report, but as it is, it only raises questions about possible overall mechanism.

5) In the discussion, the claims about perceived evolutionary trajectories of these microbes are overstated (line 235, 254, 261, 276) and the authors should either clarify or moderate these assertions.

We cannot make any decision about publication until we have seen the revised manuscript and your response to the reviewers' comments. Your revised manuscript is also likely to be sent to reviewers for further evaluation.

Sincerely,

Peggy A. Cotter

Guest Editor

PLOS Pathogens

D. Scott Samuels

Section Editor

PLOS Pathogens

Michael Malim

Editor-in-Chief

PLOS Pathogens

orcid.org/0000-0002-7699-2064

Reviewer's Responses to Questions

**Part I - Summary**

Reviewer #1: Killing other microbes is a widespread trait among bacterial pathogens. However, the effect of bacteria-bacteria killing on the host and the trajectory of an infection are little understood. Here, Virgo and colleagues use the zebra fish as a host model for a bacterial infection with three different bacterial pathogens and make three remarkable observations: i) T6SS-mediated killing of E. coli results in decreased host survival after infection with two pathogenic species ii) T6SS-independent killing of E. coli does not affect host survival after infection with a non-pathogenic species and iii) inhibition of the host immune system during bacterial killing has the opposite effect on host survival after infection with the one or the other pathogenic species.

In my opinion, the text is very well written, the data is presented very nicely in the figures and controls are included.

I fully support the authors' efforts in performing such in vivo experiments with all the challenges and complexity that come along with them. I think this is very much needed and the results are of broad interest to groups interested in bacterial killing and pathogenesis.

I do think though that the complexity of the model and the comparison across species requires careful interpretation of the data. The authors are probably well aware that there are a few loose ends and I do not want to request a long list of additional experiments considering the solid existing data and the journal this is submitted to. I please ask the authors to take the comments below into consideration during the revisions.

Reviewer #2: Summary: The report by Virgo et al. is a creative and interesting exploration of how interbacterial dynamics alters the host inflammatory response. Although more traditional studies with mono-associations may inadvertently capture some of these dynamics, the combination of the hindbrain ventricle system and bacterial co-injections represents an innovative and useful reductive whole-organism model that will continue to be a highly useful platform for ongoing studies. Despite my overall interest and enthusiasm for this report, in my reading, I felt that several of the authors findings were either overstated, or not presented clearly enough to fully support their conclusions. Based on this, the authors should consider the addition of several experiments and controls that would enhance the strength of this report.

Overall, I believe this report merits publication, and I hope my concerns and suggestions are useful to the authors. I anticipate the editors will weigh my comments and criticisms carefully and provide advice to the authors on those which they deem most pertinent and necessary for publication within the focus of this journal.

**Part II – Major Issues: Key Experiments Required for Acceptance**

Reviewer #1: Consideration of alternative explanations:

1. As described below for the title, where do the authors know that differences between the findings with Vc and Ss infections are because of the different molecular mechanisms and not because of the different bacterial species used? To me, there is too much focus on the former interpretation and too little discussion of the latter. How about focusing more on the T6SS-mediated effect across the two pathogens and less on the comparison between T6SS and colicin-mediated killing in the overall story of the manuscript?

2. Maybe I missed this, did the authors mention the known anti-eukaryotic activity of the multiple T6SS effectors (including VgrG1) of V. cholerae? Although it seems to be as expected to the authors, to me it is unexpected that there is no phenotype of the ∆T6SS mutant on the innate immune response or host survival (l. 119ff). Even if there is no effect after infection with the V. cholerae strain alone, could this anti-eukaryotic activity still play a role after the immune system is triggered by the E. coli lysate in infections with both species? Did the authors consider injecting the zebra fish with E. coli lysate alone (corresponding to a similar number of live E. coli that were injected in the experiment), E. coli lysate together with a V. cholerae ∆T6 mutant, and E. coli lysate together with a V. cholerae WT strain? I think such experiments would very much strengthen the manuscript and help disentangle whether the observed phenotype is purely based on E. coli cell lysate, the co-occurrence of E. coli lysate in combination with V. cholerae per se (independent of the T6SS) or the co-occurrence of E. coli lysate, the pathogen V. cholerae and anti-eukaryotic activity of T6SS effectors.

Fig. 7A, I like the idea, however I think the conclusions on host health as indicated on the right of the figure are not justified with the current experimental set-up because of the use of pathogenic and non-pathogenic species for the T6SS and colicin phenotype. Further, there is no sign of inflammation shown in Figure 6, isn’t the blue line for the inflammatory response upon colin-mediated killing purely hypothetical and would require more experiments to be backed up?

Reviewer #2: Major concerns:

1) Figure 2 represents foundational finding of the paper, yet the comparison between Vibrio and Vibrio + E. coli in terms of host survival and inflammatory cytokine secretion is only indirectly made through comparison of independent experiments. Although a comparison between the results of figure 1 and 2 can be made, the authors should show a direct comparison of cytokine and survival between mono-associated Vibrio and Vibrio + E. coli to clearly demonstrate the differences in host response of this key finding. Ideally this would be a discrete independent experiment, but at the very least, a compilation of data from figure 1 and 2 into a single graph (and indicated as such).

2) The authors’ claim that continuous, but weaker bacterial T6SS mediated antagonism is more immunostimulatory due to its continuous and chronic release of E. coli DAMPs is a potentially reasonable conclusion but is only weakly supported by experimental evidence, and is somewhat of a logical leap as it stands now.

a. Injection of colicin vs. T6SS killed E. coli into the hindbrain ventricle in the absence of Vibrio, then assess survival and inflammatory cytokine response to these forms of killed bacteria (in the absence of Vibrio or A. bayli). This would demonstrate that acute responses to a killed bacteria is the same between these two mechanisms, and not due to differential DAMP release based on mechanism of bacterial killing.

3) The dexamethasone experiments are interesting and an excellent approach but are not properly controlled. Please include controls evaluating the effect of dexamethasone on Vibrio and A. bayli survival and inflammatory cytokine in the absence of E. coli. As it is, it is conceivable that dexamethasone treatment alone makes Vibrio infection pathogenic even in the absence of E. coli.

4) It is not clear that lethality in any of these models is due to the host response or a direct bacterial effect, nor is it particularly clear what role the host response is playing on bacterial grown. Although I believe it is beyond the scope of this report to experimentally address this question, the authors should indicate these unknowns and possibilities for addressing them in future studies.

**Part III – Minor Issues: Editorial and Data Presentation Modifications**

Reviewer #1: - title: I agree with there being interbacterial competition and I agree with distinct host responses and health outcomes, I struggle with this being because of “different modes” of interbacterial competition. Are the authors sure there are no other possible explanations considering that different species were used to test the different molecular mechanisms? I suggest to leave the first three words out of the title.

- abstract: for clarity, I suggest to add the name of the bacterial species with which colin-mediated bacterial killing was tested (as is already done for T6SS-mediated killing).

- for all Kaplan Meier curves with pooled data, could the authors please provide the data of the individual experiments in the supplementary data so that the consistency of the phenotype across independent experiments becomes visible?

- l. 98, although this might hold, I do think more specific work would be needed to specifically conclude on “evolutionary advantages” and suggest not to do so here. Especially with an infection model that might be a very nice scientific model system but not necessarily represent the natural route for these pathogens.

Reviewer #2: (No Response)

PLOS authors have the option to publish the peer review history of their article (what does this mean?). If published, this will include your full peer review and any attached files.

Reviewer #1: No

Reviewer #2: No

Figure Files:

Data Requirements:

Reproducibility:

To enhance the reproducibility of your results, we recommend that you deposit your laboratory protocols in protocols.io, where a protocol can be assigned its own identifier (DOI) such that it ca

---

## [Decision Letter · Decision Letter 1]

1 Jul 2024

Dear Prof Mostowy,

We are pleased to inform you that your manuscript 'Interbacterial competition elicits distinct host responses and health outcomes' has been provisionally accepted for publication in PLOS Pathogens.

Both reviewers agreed that the concerns raised were adequately addressed and that the work represents a substantial and significant advance in understanding that will be appreciated by the field. Reviewer #2, however, had one concern that can be addressed by modifying the text. This reviewer stated that in the Results section titled, "V. cholerae T6SS antagonism towards E. coli prolongs the inflammatory response and reduces host survival," it appears that the lethality of vibrio alone (either WT or deltaT6) is nearly equivalent to the lethality of E. coli + WT vibrio. This result suggests that host lethality is primarily mediated by bacterial burden and not the host response causing immune mediated pathology (supported by the dexamethasone findings of figure 4). Although figure 2 clearly demonstrates an increased inflammatory response due to T6-mediated E. coli killing, it is still not clear that this results in direct immune pathology and self-damage to the host. The reviewer believes, and I agree, that it is important to clarify how you believe the pathology is being mediated. Altering the wording in the text to clarify and avoid misinterpretation would suffice. As this should be a relatively minor change or addition, you can do it when addressing the formatting changes.

Best regards,

Peggy A. Cotter

Guest Editor

PLOS Pathogens

D. Scott Samuels

Section Editor

PLOS Pathogens

Michael Malim

Editor-in-Chief

PLOS Pathogens

orcid.org/0000-0002-7699-2064

Both reviewers agreed that their concerns were adequately addressed in the revised version and that the work represents a substantial advance in understanding that will be appreciated by the field. Reviewer #2 had once concern that can be addressed by modifying the text. That concern is:

"Results section header titled: V. cholerae T6SS antagonism towards E. coli prolongs the inflammatory response and reduces host survival

Looking at the survival curves in figure 1, it appears that the lethality of vibrio alone (either WT or deltaT6) is nearly equivalent to the lethality of E. coli + WT vibrio - is this correct or am I misinterpreting the data?

I point this out because it suggests that host lethality is primarily mediated by bacterial burden and not the host response causing immune mediated pathology (supported by the dexamethasone findings of figure 4). Although figure 2 clearly demonstrates an increased inflammatory response due to T6 mediated e. coli killing, it is still not clear that this results in direct immune pathology and self-damage to the host. While I don't think you are necessarily intending to suggest this, I do think it's potentially important to clarify on how you believe the pathology is being mediated.

I therefore recommend that the authors either make a direct comparison of WT vibrio vs WT vibrio + e.coli (this would need to be an independent experiment), or alter the wording in the text to clarify and avoid misinterpretation."

I recommend modifying the text as suggested by the reviewer.

Reviewer Comments (if any, and for reference):

Reviewer's Responses to Questions

**Part I - Summary**

Reviewer #1: I thank the authors for their work on the revised manuscript. All my comments were addressed to my satisfaction.

Reviewer #2: This revised version addresses all my concerns and I believe it presents the data with significantly more transparency and clarity.

Although I feel that the dynamics of vibrio sensitivity to interbacterial competition vs the host response remain somewhat obscure, this is a complex biologic system and a the expectation for a completely reductive answer to these questions is unlikely and unreasonable. The authors present well-thought through experiments that do an excellent job of investigating these dynamics.

**Part II – Major Issues: Key Experiments Required for Acceptance**

Reviewer #1: (No Response)

Reviewer #2: No major concerns.

**Part III – Minor Issues: Editorial and Data Presentation Modifications**

Reviewer #1: (No Response)

Reviewer #2: Results section header titled: V. cholerae T6SS antagonism towards E. coli prolongs the inflammatory response and reduces host survival

Looking at the survival curves in figure 1, it appears that the lethality of vibrio alone (either WT or deltaT6) is nearly equivalent to the lethality of E. coli + WT vibrio - is this correct or am I misinterpreting the data?

I point this out because it suggests that host lethality is primarily mediated by bacterial burden and not the host response causing immune mediated pathology (supported by the dexamethasone findings of figure 4). Although figure 2 clearly demonstrates an increased inflammatory response due to T6 mediated e. coli killing, it is still not clear that this results in direct immune pathology and self-damage to the host. While I don't think you are necessarily intending to suggest this, I do think it's potentially important to clarify on how you believe the pathology is being mediated.

I therefore recommend that the authors either make a direct comparison of WT vibrio vs WT vibrio + e.coli (this would need to be an independent experiment), or alter the wording in the text to clarify and avoid misinterpretation.

PLOS authors have the option to publish the peer review history of their article (what does this mean?). If published, this will include your full peer review and any attached files.

Reviewer #1: No

Reviewer #2: No

---

## [Editor Report · Acceptance letter]

12 Jul 2024

Dear Prof Mostowy,

We are delighted to inform you that your manuscript, "Use of zebrafish to identify host responses specific to Type VI Secretion System mediated interbacterial antagonism," has been formally accepted for publication in PLOS Pathogens.

Best regards,

Michael Malim

Editor-in-Chief

PLOS Pathogens

orcid.org/0000-0002-7699-2064